# Obesity-induced senescent macrophages activate a fibrotic transcriptional program in adipocyte progenitors

Nabil Rabhi[1], Kathleen Desevin[1], Anna C Belkina[2,3], Andrew Tilston-Lunel[1], Xaralabos Varelas[1], Matthew D Layne[1], Stephen R Farmer[1]

Adipose tissue fibrosis is regulated by the chronic and progressive metabolic imbalance caused by differences in caloric intake and energy expenditure. By exploring the cellular heterogeneity within fibrotic adipose tissue, we demonstrate that early adipocyte progenitor cells expressing both platelet-derived growth factor receptor (PDGFR) $\alpha$ and $\beta$ are the major contributors to extracellular matrix deposition. We show that the fibrotic program is promoted by senescent macrophages. These macrophages were enriched in the fibrotic stroma and exhibit a distinct expression profile. Furthermore, we demonstrate that these cells display a blunted phagocytotic capacity and acquire a senescence-associated secretory phenotype. Finally, we determined that osteopontin, which was expressed by senescent macrophages in the fibrotic environment promoted progenitor cell proliferation, fibrotic gene expression, and inhibited adipogenesis. Our work reveals that obesity promotes macrophage senescence and provides a conceptual framework for the discovery of rational therapeutic targets for metabolic and inflammatory disease associated with obesity.

## Introduction

White adipose tissue (WAT) exhibits remarkable plasticity and plays a central role in regulating systemic energy homeostasis. During nutrient excess, WAT stores surplus energy in the form of triglycerides, and during nutrient deficits, WAT releases energy to other tissues through lipolysis (Farmer, 2009). In response to chronic energy excess, WAT undergoes a dynamic remodeling response involving increases in adipocytes size (hypertrophy) and number (hyperplasia) (Sun et al, 2011). In addition, in response to obesity various stromal-vascular cells (SVCs) including progenitor and immune cells undergo numerical and functional changes leading to WAT fibrosis and inflammation, thereby exacerbating metabolic dysfunction (Sun et al, 2011, 2013). The detailed knowledge of the contribution of each cell type to WAT remodeling and the

mechanisms governing cell to cell communication in a fibrotic stroma are poorly understood. SVC are a dynamic and complex assortment of resident immune cells, vascular cells, mesenchymal stromal cells (MSCs), and preadipocytes that change with WAT development and remodeling (Eto et al, 2009). Platelet-derived growth factor receptors (PDGFRs) $\alpha$ and $\beta$ mark the adipose progenitor cells in the stromal-vascular niche (Berry & Rodeheffer, 2013; Lee et al, 2014; Vishvanath et al, 2016). Indeed, multiple studies have reported that the PDGFR$\alpha$ lineage generates most of the adipocytes in response to adipogenic stimuli (Berry & Rodeheffer, 2013; Lee et al, 2014). In contrast, studies of the PDGFR$\beta$ lineage reached different conclusions (Vishvanath et al, 2016). Recent studies using PDGFR$\alpha$ and PDGFR$\beta$ mosaic lineage labeling showed that each lineage makes distinct contributions to adipocyte differentiation (Sun et al, 2020). In addition, interstitial cells expressing dipeptidyl-peptidase 4 (DPP4) were proposed to be a renewable source of preadipocytes in response to various stimuli (Merrick et al, 2019). However, the overlap and the interrelationships among these and other described cell populations and their contribution to WAT fibrosis are incompletely understood. The other hallmark of obesity is inflammation sustained by the infiltration of immune cell populations including macrophages, dendritic cells, mast cells, neutrophils, B cells, and T cells (Ouchi et al, 2011). Macrophages are the most abundant leukocyte in WAT of both mice and humans and contribute to insulin resistance during obesity by promoting a proinflammatory phenotype (Wynn et al, 2013). They aggregate in WAT to form crown-like structures (CLSs) and adopt a metabolically activated phenotype to eliminate dead adipocytes (Murano et al, 2008). CD9[+] macrophages in particular, which are also localized in CLS, exhibit a metabolic activation transcriptional signature and are responsible for the inflammatory signature of obese adipose tissue (Hill et al, 2018). A novel macrophage subset named lipid-associated macrophage expressing the lipid receptor Trem2 also resides in the CLS (Jaitin et al, 2019). Deletion of Trem2 in bone marrow cells led to deterioration of the metabolic outcomes in obese mice. However, it is currently unknown whether macrophage functional heterogeneity contributes to WAT fibrosis and how these cells communicate with adipocyte progenitor cell differentiation pathways. Here, we characterize the cellular landscape of fibrotic stroma in WAT and dissect

[1]Department of Biochemistry, Boston University School of Medicine, Boston, MA, USA   [2]Flow Cytometry Core Facility, Boston University School of Medicine, Boston, MA, USA   [3]Department of Pathology and Laboratory Medicine, Boston University School of Medicine, Boston, MA, USA

Correspondence: rabhi@bu.edu; mlayne@bu.edu; sfarmer@bu.edu

how prolonged diet-induced obesity reshapes the microenvironment. We use single-cell RNA sequencing to provide a comprehensive atlas of cell types implicated in WAT fibrosis. We uncovered that early progenitors expressing both PDGFRα and β are the major contributors to ECM deposition. Multi-lineage modelling of ligand and receptor interactions between ECM producing cells and stromal cell populations revealed activity of several pro-fibrogenic signaling molecules. We identified a CD9[+] subpopulation of macrophages, which expands in fibrotic stroma, as a significant source of profibrogenic and proinflammatory factors. Functional analysis demonstrated that CD9[+] macrophages are senescent and exhibit a blunted phagocytotic capacity. Osteopontin expressed by senescent macrophages in the fibrotic stroma promoted progenitor cell proliferation, fibrotic gene expression, and inhibited adipogenesis. Our studies identified new mechanisms that may explain the impaired immune function in the obese patient and obesity-associated metabolic disorders and inflammatory pathologies.

# Results

To identify cellular changes occurring in the fibrotic WAT stroma, we applied single-cell RNA sequencing (scRNAseq) to the stromal fraction isolated from epididymal WAT (eWAT) of C57BL/6J male mice fed either low-fat diet (LFD) or high-fat diet (HFD) starting at 6 wk of age for 24 wk. Unsupervised clustering of gene expression profiles identified 17 cell types (Fig 1A). Analysis of integrated data identified several cell populations, including committed preadipocytes (PreAd), four subpopulations of mesenchymal stromal cells (MSC1-4), mesothelial cells (Meso), endothelial cells (Endo), immune cells (including macrophages, natural killer cells, B cells, T cells, and dendritic cells) and a subpopulation of cells expressing markers of hematopoietic stem cells. Consistent with previous reports (Vijay et al, 2020), comparisons between LFD and HFD showed a marked increase in the relative number of endothelial cells, immune cells and MSC (Figs 1B and S1A and B). Analysis of the top 10 genes expressed in MSCs reveals five distinct clusters comprising MSC1-4 and preadipocytes (Fig 1C and D) Progenitor cell markers including Cd34, Ly6a (Sca1) and Pdgfrα were expressed in both MSC1 and MSC3, whereas Pdgfrβ was primarily expressed by the MSC1 cluster (Fig 1C) (Berry & Rodeheffer, 2013; Rodeheffer et al, 2008). Interstitial cell markers such as Pi16 and Dpp4 were expressed in MSC3 and to a lower level in MSC1 (Fig 1C) (Merrick et al, 2019). Markers of adipogenesis regulators (e.g., F3, Gdf10, Meox2, and Agt) were scattered through all MSC populations (Fig 1C) (Schwalie et al, 2018). Committed preadipocyte markers including Fabp4, Plin2, and Car3 were also found in MSCs and in committed preadipocytes; only adiponectin specifically marked the latter (Fig 1C and D).

## Progenitor cells are the major ECM producing cells

Pseudotime analysis using markers from MSCs and preadipocytes (Fig 2A and Table S1) predicted that MSC3 cells have two trajectories, either giving rise to MSC2 (fate A) or preadipocytes (fate B) with MSC1 and MSC4 appearing to represent an intermediary cell type (Figs 2A and S2A and B). MSC4 cells appear to be a mix of two subpopulations that arise from MSC1, each of which can undergo either fate A or fate B. Further analysis of temporal gene expression identified markers that are down-regulated early during MSC3 commitment to both fates such as Pi16, Sema3C, and Mfap5, others are re-expressed when the cells undergo fate B (e.g., Socs3 and Atf3). Interestingly, the expression of some intermediary cell markers such as Cldnl5, Gpihbp1, and Kdr is maintained in cells undergoing fate B, whereas repressed when the cells differentiate to preadipocytes (Figs 2B and S2C). Altogether, these data suggest that interstitial cells (MSC3) give rise to an intermediary cell population (MSC1) that has two distinct fates, one of which gives rise to adipocytes. We next analyzed gene ontology (Go) terms associated with genes up-regulated when MSC clusters are compared with each other. Marker genes of MSC1 cluster were associated with collagen organization, ECM structure and organization, and cartilage and connective tissue development (Fig S2D). MSC2 cluster markers were associated with antigen presentation, endothelium development, and endothelial cell differentiation suggesting that fate B may provide a source of endothelial cells (Fig S2E). Go terms enriched in MSC3 cluster markers were primarily associated with ribosomal activity and biogenesis and chondrocytes differentiation and cartilage development suggesting that those cells have a proliferative capacity and are localized within the interstitium (Fig S2F). MSC4 cluster markers were associated with GTPase signaling, protein dephosphorylation and apical junction formation suggesting that those pathways may be important for the regulation of cell fate decisions (Fig S2G). Interestingly, only MSC1 Go analysis showed an increase in ECM pathways. Indeed, further analysis showed that ECM genes such as Col6a3, Col3a1 and Col1a2 were primarily expressed by early progenitors MSC1 and to a lesser extent by MSC3 (Fig 2C). Furthermore, analysis of the composition of MSC1 markers in the obese versus lean states showed a marked increase in ECM genes (Figs 2D and S2H). In addition, enriched Go terms indicated that HFD led to an increase in pathways associated with collagens fibril and ECM organization, inflammatory response and metabolic processes (Fig 2E). Altogether, our data suggest that MSC1 are the major ECM producing cells that also give rise to preadipocytes.

## Adipocyte progenitor cells express both PDGFRα and PDGFRβ and expand under HFD

Flow cytometry analysis of CD34 cells revealed that PDGFRα and PDGFRβ account for 58% of the total CD34 population under LFD. Their number increased to 90% in eWAT from HFD mice (Fig 3A). PDGFRα and CD34 double positive cells were the second most abundant population followed by CD34[+] PDGFRα and β negative cells, whereas CD34[+] and PDGFRβ[+] represented fewer than 2% of the total CD34[+] cells. HFD led to a significant decrease in both PDGFRα[+] and double negative cells, whereas it had no effect on PDGFRβ[+] cells number (Figs 3A and S3A). We next assayed the differentiation capacity of each population. PDGFRα[+] and PDGFRβ[+] double positive cells and PDGFRα[+] positive cells undergo a robust adipocyte differentiation when treated with a standard adipogenic cocktail (Figs 3B and S3B and C). Interestingly, cells negative for PDGFRα, PDGFRβ, and CD34 spontaneously differentiated to adipocytes after 48 h of seeding, suggesting those cells represent committed preadipocytes

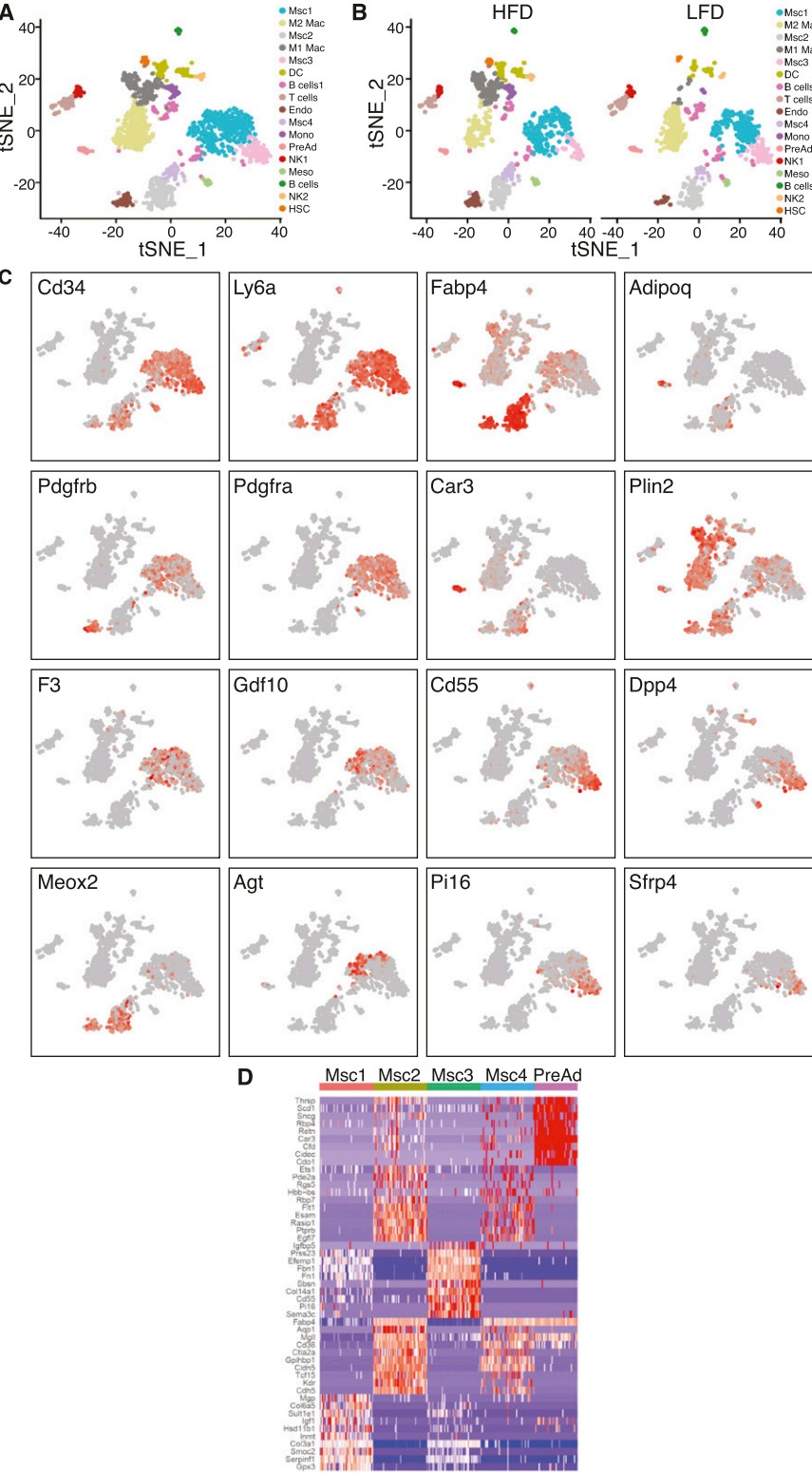

**Figure 1. Characterization of the fibrotic stromal microenvironment.**
**(A)** t-SNE plots of eWAT SVF populations identified by single cell from combined dataset of 24-wk low-fat diet and high-fat diet feed mice, which identified 17 clusters. **(B)** Side-by-side t-SNE plots of eWAT SVF populations from 24-wk low-fat diet and high-fat diet feed mice. **(C)** t-SNE expression plots of differentially expressed genes displaying select markers. **(D)** Heat map of top 10 marker genes differentially expressed in mesenchymal stem cells and preadipocytes.

in scRNAseq cluster (Fig S3B). Both PDGFRβ⁺ cells and cells positive for only CD34 did not exhibit adipogenic potential suggesting that both of these populations could be progenitors for other cell subtypes. Last, we stained sections of eWAT from mice under LFD and HFD for PDGFRα and PDGFRβ. Consistent with the flow cytometry results, immunostaining of eWAT showed that PDGFRα

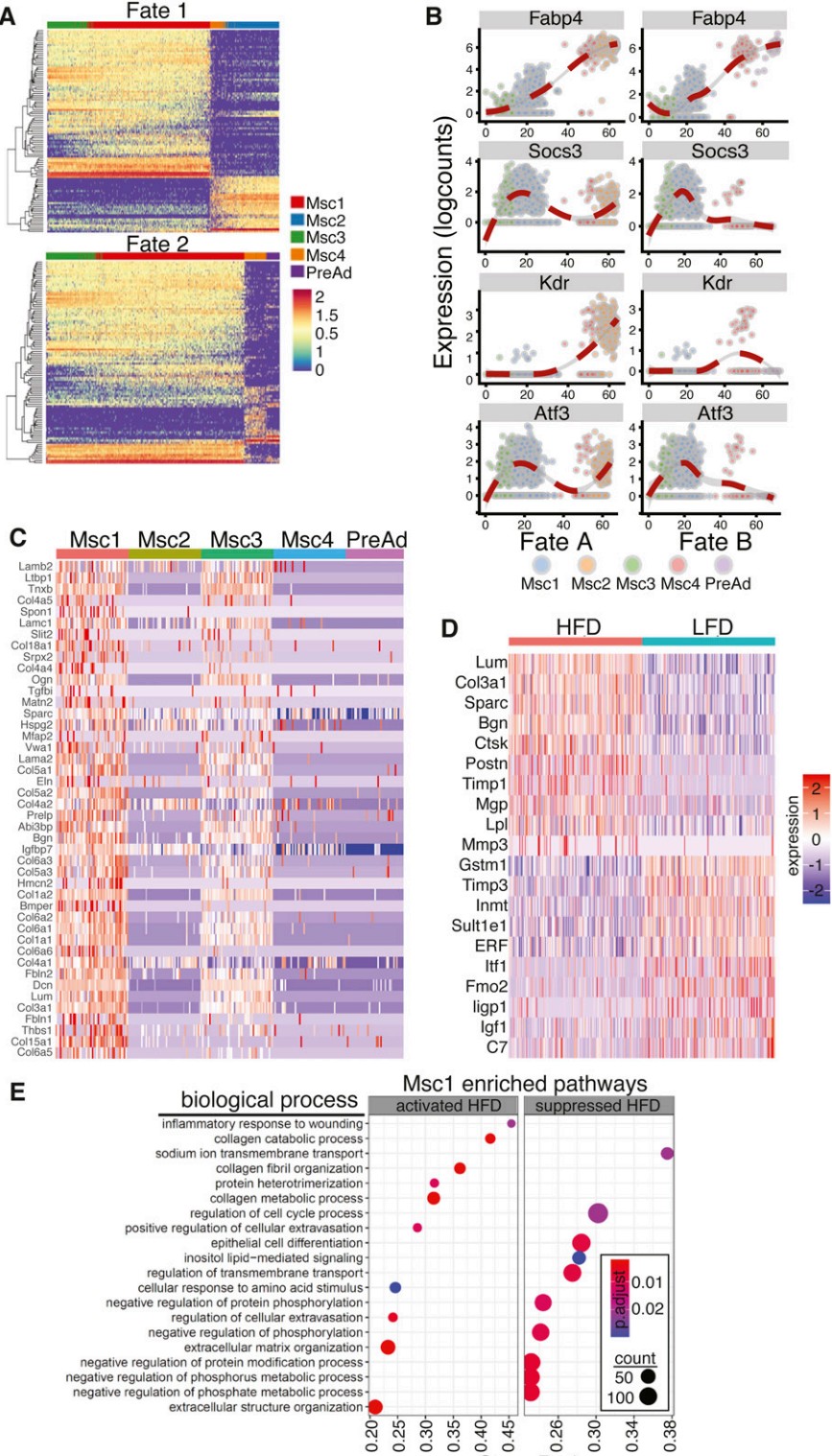

**Figure 2. Progenitor cells produce ECM.**
**(A)** Heat map of the top 250 genes that are associated with each lineage identified. **(B)** Expression profiles of select genes during both lineages identified; dots represent a cell and line represents the loess regression. **(C)** Heat map of ECM genes differentially expressed in mesenchymal stem cells and preadipocytes. **(D)** Heat map of ECM genes differentially expressed between low-fat diet and high-fat diet in MSC1 population. **(E)** Dot plot of gene ontology (biological process) of activated and suppressed pathways by high-fat diet in MSC1 cluster.

and PDGFRβ colocalized under LFD and the double positive cell number expanded in fibrotic tissue (Fig 3B and C). Gene expression profiling of sorted CD34⁺, PDGFRα⁺, and PDGFRβ⁺ cells showed that HFD promoted expression of ECM genes including Col1a1, Col3a1,

and Timp1 (Fig 3D). Altogether, these results suggest that PDGFRα/PDGFRβ double positive cells represent early progenitors. The loss of PDGFRβ is important for the progenitors to commit to the adipocyte lineage, whereas the loss of PDGFRα facilitates non-

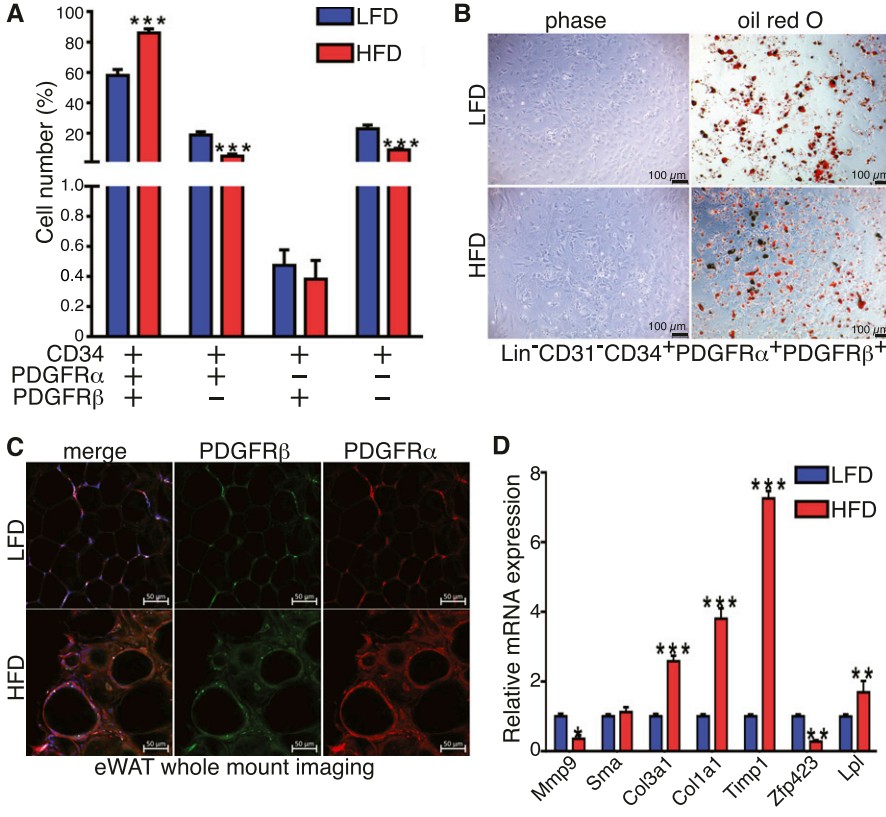

**Figure 3.  Early progenitor cells express both platelet-derived growth factor receptor (PDGFR)α and PDGFRβ and expand in mice fed a high-fat diet (HFD).**
**(A)** Quantification of frequency of Lin⁻: CD31⁻: CD34⁺ and either PDGFRα⁺ or PDGFRβ⁺ or double positive cells in the eWAT depot from 24-wk low-fat diet (LFD) and HFD feed mice performed by flow cytometry (n = 8). **(B)** Representative images of Oil Red O staining of in vitro differentiated Lin⁻ : CD31⁻ : CD34⁺ : PDGFRα⁺ and PDGFRβ⁺ sorted cells form eWAT depot in 24-wk LFD and HFD feed mice, scale bars 100 µm. **(C)** Representative image of whole mount of PDGFRα and PDGFRβ immunofluorescent co-staining, scale bars 50 µm. **(D)** Real-time PCR analysis of select genes in Lin⁻: CD31⁻ : CD34⁺ : PDGFRα⁺ and PDGFRβ⁺ sorted cells form eWAT depot in 24-wk LFD and HFD feed mice (n = 4). All values are expressed as means ± SEM; statistical test *P < 0.05, **P < 0.01, and ***P < 0.001.

adipogenic lineages possibly endothelial. Moreover, in an obese state progenitor cells became activated and enhanced ECM production contributing to tissue fibrosis.

### CD9⁺ macrophages communicate with PDGFRα and PDGFRβ progenitor cells through the production of osteopontin

Cellular communication has been implicated in tissue homeostasis and disease (Kahn et al, 2019). Adipose tissue is organized into functional niches where cells communicate via physical interactions, metabolite exchange, and via ligand–receptor interactions, forming cellular networks (Estève et al, 2019). To identity how cellular changes in the fibrotic stroma affect ECM production by MSC1, we computationally analyzed intercellular communication between all identified clusters and MSC1. We used NicheNet to predict potential ligands secreted by each identified cluster cell population (sender cells) that can influence regulation of differentially expressed genes in the MSC1 population (receiver cell) (Browaeys et al, 2020). Ligands secreted by "sender" cells were first matched with potential receptors expressed in MSC1 (Fig S4A). The receptors for the top predicted ligands included Bmpr2, Bmpr1a, Acvr1, Tgfbr1, Tgfbr2, Tgfbr3, Cd44, Itgb5, Itga5, Tnfrsf1a, Fgfr1, Nrp2, Il6ra, Adipor1, and Il1ra were all up-regulated in the "receiver" cell. Only ligands significantly up-regulated by HFD feeding in "sender" cells were selected for further analysis (Fig 4A). The resulting list of ligands was then incorporated with intracellular signaling up-regulated by HFD in the receiver cell

population (Fig S4B). Interestingly, many ligands identified affected pathways related to ECM and inflammatory genes. Finally, ligand activities were ranked using a NicheNet-generated Pearson correlation coefficient indicating the correlation between the target genes of a given ligand and the list of differentially expressed target genes in the "receiver" cell (Fig 4B). Among the identified ligands were Tgfb1, Bmp2, Tnf, Il6, and Il1b, all of which have been previously implicated in adipose tissue fibrosis. Differential gene expression showed that only Spp1 (OPN; Osteopontin) was exclusively up-regulated by HFD (Figs 4C and S4C). Gene expression and immunostaining showed that OPN was secreted by a subpopulation of CD9⁺ macrophages appearing in the obese state (Fig 4C–E). This macrophage population also expressed Tgfb1 and Tnfa (Fig S4C). These results demonstrate that a new population of CD9-positive macrophages emerge under HFD and likely alter the gene expression profile of MSC1 cells through the secretion of several proteins most notably Opn.

### Accumulation of senescent CD9⁺ macrophages in HFD WAT

To further characterize the macrophage populations, we first re-clustered all macrophage populations identified by scRNAseq resulting in five distinct subpopulations (Fig 5A). Interestingly, three populations were enriched under HFD, whereas only two populations were identified under LFD. Differential expression and annotation of these macrophages uncovered that two of the subpopulations identified under HFD were either M1-like or M2-like

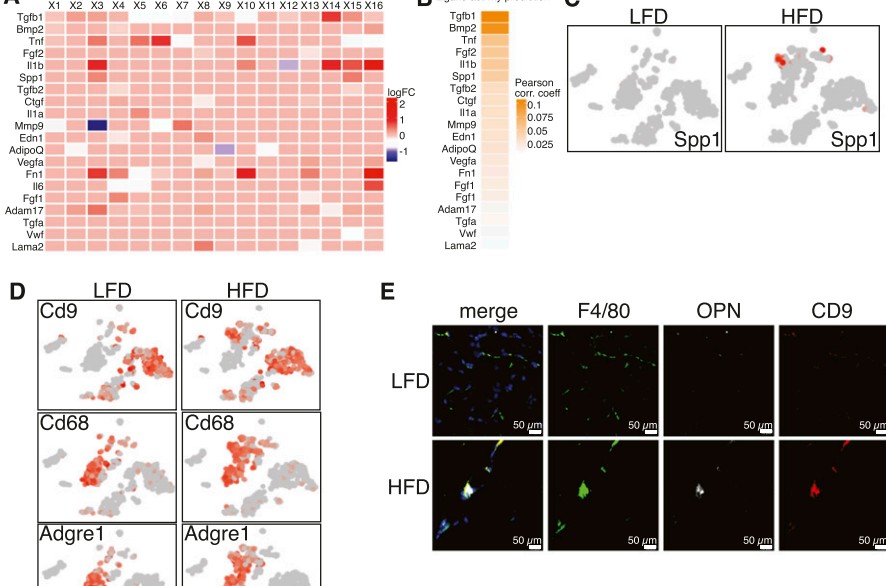

**Figure 4. CD9+ macrophages communicate with platelet-derived growth factor receptor α and platelet-derived growth factor receptor β progenitor cells through the secretion of osteopontin.**
**(A)** "Sender" cells differentially express potential pan-ligands a cross each cluster between low-fat diet (LFD) and high-fat diet (HFD) eWAT. Ligand-cell pairings indicate cell types in which the corresponding ligand is significantly differentially five expressed in HFD with adjusted $P < 0.05$. Average natural log fold change (ratio of HFD to LFD expression) is shown in the blue-red color scale. **(B)** Ligand activity prediction ranked by Pearson correlation coefficient (orange color scale). **(C)** t-SNE expression plots of osteopontin (Spp1). **(D)** t-SNE expression plots of select macrophages markers and Cd9. **(E)** Representative image of F4/80, OPN, and CD9 immunofluorescent co-staining in 24-wk LFD and HFD feed mice, scale bars 50 μm.

metabolically activated macrophages (Figs 5A and S5A). Whereas M1-like and M2-like macrophages Go terms showed an enrichment of canonical macrophages polarization pathways, Go terms for activated macrophages were enriched with ribosomal biosynthesis and assembly suggesting that HFD has a strong effect on protein translation rather than gene expression (Fig S5B–E). Go terms enriched in CD9 macrophages were primarily associated with lipid metabolism and extracellular organization (Fig 5B). Previous work showed that CD11c expression is a hallmark of inflammatory phenotype in adipose tissue macrophages (Lumeng et al, 2007). We designed a flow cytometry panel that distinguishes between populations enriched in CD11b, CD11c, and CD9 macrophages. In concordance with previous reports, HFD induced a shift toward a CD11c+ phenotype (Figs 5C and S5F) (Lumeng et al, 2007). To our surprise, we found a strong enrichment of CD11c+/CD9+ macrophages, whereas CD11c+ and CD11c− macrophages were reduced by HFD (Figs 5C and D and S5F). Consistent with our scRNAseq data, gene expression profiling of sorted macrophages showed that the CD9+ population express markers including Cd9, Spp1, Lpl, and Cd36 and the cytokines Il1b and Cxcl1 (Fig 5E). These results indicate that CD9+ macrophages are a distinct macrophage population enriched in fibrotic stroma.

Evaluation of the phagocytic capacity of the macrophages showed that it was severely blunted by HFD (Figs 6A and S6A and B). Because loss of function and secretion of proteins such as OPN and IL1b are associated with senescent cells, we performed senescence-associated β galactosidase (SA-βgal) staining of freshly isolated eWAT sections from mice under LFD or HFD for 24 wk. All cells in CLSs were positive for SA-βgal staining suggesting HFD led to senescence of macrophages (Fig 6B). To further characterize the senescent macrophages in direct response to obesity, eWAT sections from mice fed either LFD or HFD for 8, 20, and 34 wk were identified with the

macrophage marker Mac2, and co-stained with antibodies against p16 and p21 (Fig 6C). Mac2+ cells in the CLS of HFD sections expressed both p16 and p21 (Fig 6C). These cells were not observed in sections from mice on LFD. Altogether, these experiments demonstrate that the CD9+ macrophages that accumulate during obesity are senescent cells that exhibit less phagocytic capacity and have a high level of secretory capacity.

## Osteopontin produced by CD9+ macrophages cooperates with PDGF-BB to stimulate progenitor cell proliferation and inhibit adipogenesis

We next evaluated the role of OPN in the regulation of progenitor cells. CD34+ cells were isolated from CD45 depleted SVC fraction using anti-CD34+ Miltenyi beads and cultured in black, flat-bottom 96-well plates for proliferation, and differentiation assays. Because both Pdgfrα and Pdgfrβ were highly expressed in the CD34+ population, we also treated the cells with PDGF-BB; which activates all Pdgfrα and Pdgfrβ homo and heterodimers (Chen PH et al, 2013; Donovan J et al, 2013); in combination with OPN. We found that treatment with OPN or PDGF-BB alone augmented the proliferation potential of progenitor cells. This effect was further enhanced when cells were treated with a combination of both proteins, suggesting a synergic action of OPN and PDGF-BB (Figs 7A and S7). Interestingly, OPN treatment increased the expression of ECM genes including Aebp1/ACLP, Col1a1, and Col3a1 and MSC1 markers such as Col15a1, Pdgfra, and Pi16 (Fig 7B). Combined treatment with OPN and PDGF-BB enhanced expression of these markers as well as the proliferation marker Mki67 while reducing cell cycle inhibitors P16 and P19 expression (Fig 7C). Moreover, whereas OPN and PDGF-BB treatment alone had a slight effect on the adipogenic capacity of progenitor cells, combined treatment completely blocked the adipogenesis

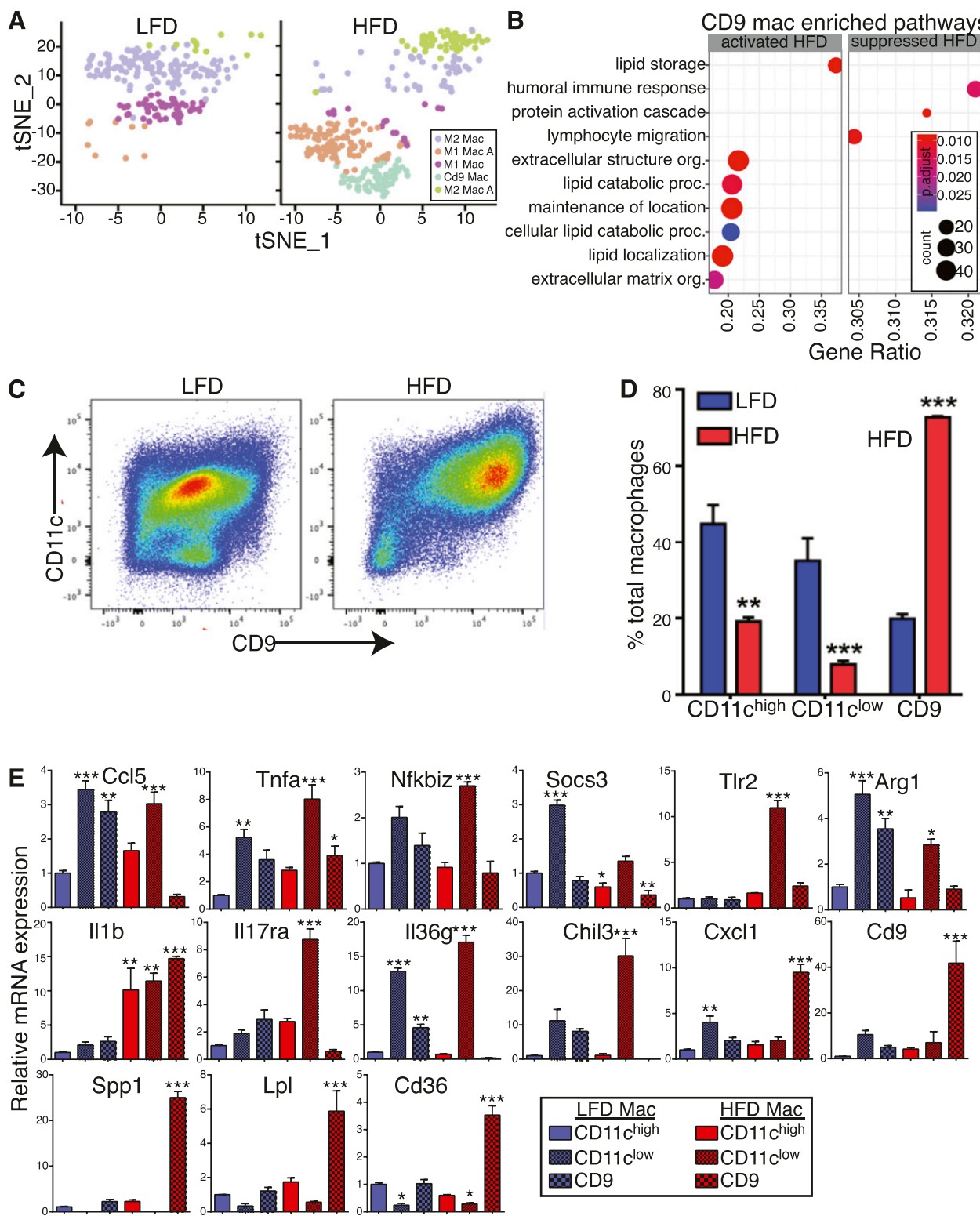

**Figure 5. Accumulation of CD9+ macrophages in high-fat diet (HFD) white adipose tissue.**
**(A)** Side-by-side t-SNE plots of macrophages. **(B)** Dot plot of gene ontology (biological process) of activated and suppressed pathways by HFD in CD9 macrophages. **(C)** Representative high scatter dot plot images of flow cytometry showing the appearance of CD11c, CD9 population in HFD. **(D)** Quantification of frequency of macrophages populations in the eWAT depot from 24-wk low-fat diet and HFD feed mice performed by flow cytometry (n = 8). **(E)** Real-time PCR analysis of select genes in sorted macrophages form eWAT depot in 24-wk low-fat diet and HFD feed mice.

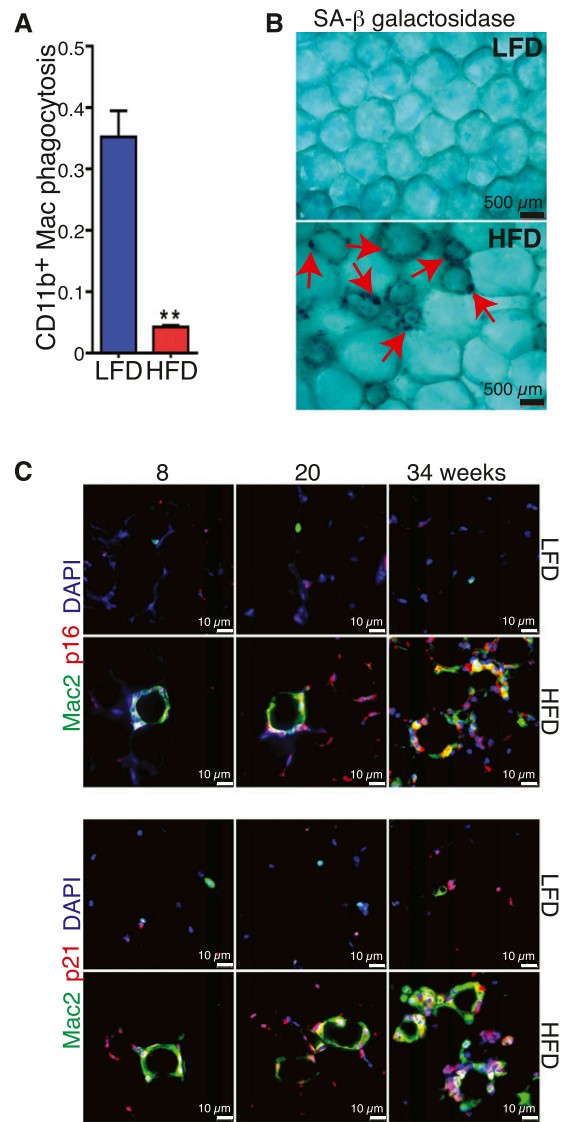

**Figure 6. CD9+ macrophages are senescent cells.**
**(A)** Quantification of macrophages in the eWAT depot from 24-wk low-fat diet (LFD) and high-fat diet (HFD) feed mice phagocytic capacity measured by flow cytometry (n = 8). **(B)** Senescence-associated (SA) b gal staining in fresh isolated eWAT from 24-wk LFD and HFD feed mice, scale bars 500 $\mu$m. **(C)** Representative image of Mac-2, P16, and P21 immunofluorescent co-staining in 8, 20 and 34-wk LFD and HFD feed mice, scale bars 10 $\mu$m. Data are presented as mean ± SEM. $P$-values*$P$ < 0.05, **$P$ < 0.01, ***$P$ < 0.001.

induced by the adipogenic cocktail (Fig 7D). Finally, whole mount histological examination of eWAT revealed a substantial infiltration of Opn+ cells intercalated between mature adipocytes in HFD mice (Fig 7E). Close examination of the PDGFR$\alpha$ and PDGFR$\beta$ co-expression showed an accumulation of double positive cells in the eWAT from HFD fed mice (Fig 7F). These results demonstrate that the synergic action of OPN and PDGF-BB led to increased proliferative capacity, inhibited adipogenesis, and activated fibrogenic programs within the progenitor cells mimicking the effect of HFD in vivo. In addition, OPN and PDGF-BB co-treatment maintained the progenitor phenotype of the cells contributing to

accumulation of non-differentiated ECM-producing progenitors in eWAT from obese mice.

## Discussion

Here, we applied scRNAseq to delineate dynamic changes of fibrotic WAT microenvironment and define the fibrotic stroma cellular crosstalk perturbed by diet-induced obesity. We show that obesity leads to the accumulation of $\beta$gal+, p16+, p21+ senescent macrophages. Flow cytometry analysis revealed that CD9+ macrophages are the major macrophage population in the fibrotic stroma. Previous work showed that those cells are recruited from the bone marrow and proposed as a new macrophage subtype (Hill et al, 2018). Here, we demonstrate that these CD9+ macrophages are senescent and lose their phagocytic capacity, which is inconsistent with their purported role to clear dead adipocytes in obese WAT. It is possible though that they transform from a phagocytic to a senescent phenotype as lipid deposition progresses. Macrophage senescence was shown to occur in aging mice (Liu et al, 2019). Moreover, foamy macrophages that accumulate in atherosclerotic plaques also exhibit a senescent profile (Childs et al, 2016). The CD9+ macrophages we describe here share some features of senescent foamy macrophages although we did not observe any accumulation of lipids, a hallmark of an atherosclerotic macrophage. PPAR$\gamma$ activation by lipids could be one of the mechanisms promoting the appearance of the senescent phenotype in recruited macrophages. CD9+ macrophages have been shown to accumulate and contribute to fibrosis in other tissues such as liver and lung in both human and mice suggesting that those cells maybe a hallmark of chronic scared/fibrotic tissue (Taylor et al, 2000; Ramachandran et al, 2019; Sommerfeld et al, 2019). Whether the senescent phenotype is promoted by chronic low-grade inflammation or lipid spillover during obesity is still unknown. Further studies are needed to investigate mechanisms leading to macrophages senescence.

The unbiased classification and characterization from the single-cell analysis provide distinct phenotypic profiles that were identified through examination of multiple progenitor surface markers. We identified five mesenchymal cell groups including a cluster of committed preadipocytes that spontaneously differentiated to adipocytes in vitro. Our study is consistent with previous investigations that described analogous cell populations in both humans and mice (Burl et al, 2018; Hepler et al, 2018; Schwalie et al, 2018; Merrick et al, 2019). Consistent with Merrick et al (2019), we found that the Pi16+ group represents early progenitors. This cell population gives rise to a second distinct group of cells (MSC1), which exhibited a greater proliferative capacity. The progression involves among other changes, a down-regulation of Col14a and up-regulation of Col15a, Pdgfr$\alpha$, and Pdgfr$\beta$. The fact that both Pdgfr$\alpha$ and Pdgfr$\beta$ are expressed in the MSC1 population suggests that both Pdgfr$\alpha$ and Pdgfr$\beta$ lineage tracing models mark to the same cell population (Sun et al, 2020). Recent investigations using Pdgfr mosaic lineage labeling mice support those findings by showing that Pdgfr$\alpha$ and Pdgfr$\beta$ expression are important regulators of adipocyte differentiation and down-regulation of Pdgf signaling is critical for adipogenesis (Sun et al, 2020). Our pseudotime analysis revealed that down-regulation of Pdgfr$\beta$ and Cd34

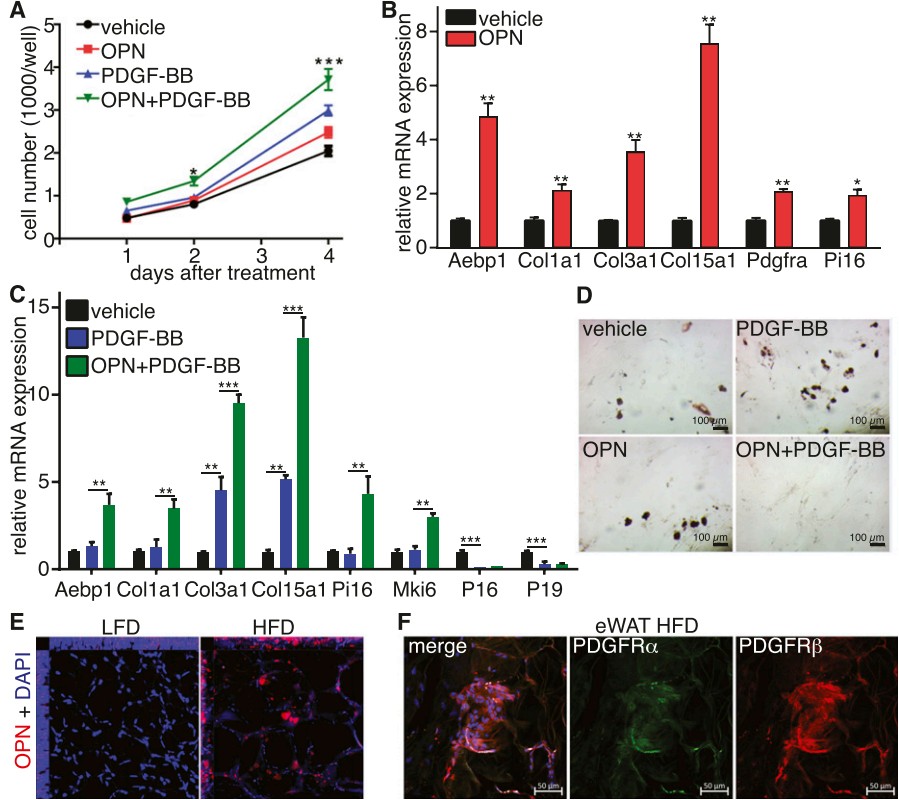

**Figure 7. Osteopontin secreted by CD9⁺ macrophages cooperate with PDGF-BB to stimulate progenitor cell proliferation and inhibit adipogenesis.**
**(A)** Quantification of cellular proliferation assay (n = 6). **(B)** Real-time PCR analysis of select genes in sorted progenitor cells treated with OPN or vehicle for 2 d. **(C)** Real-time PCR analysis of select genes in sorted progenitor cells treated with PDGF-BB alone, combined with OPN or vehicle for 2 d (n = 4). **(D)** Representative images of Oil Red O staining of in vitro differentiated progenitor cells treated with OPN, PDGF-BB, a combination of the two or vehicle for 2 d (n = 4), scale bars 100 μm. **(E)** Orthogonal view of z-stack images of whole mount immunostaining of OPN. **(F)** Representative image of whole mount of platelet-derived growth factor receptor α and platelet-derived growth factor receptor β immunofluorescent co-staining in eWAT for 24-wk high-fat diet feed mice, scale bars 50 μm. Data are presented as mean ± SEM. P-values. *P < 0.05, **P < 0.01, ***P < 0.001.

is important for MSC1 population to commit to the adipogenic lineage. Indeed, stromal cells deficient in Pdgfrs and CD34 spontaneously differentiated into adipocytes in vitro. Re-expression of Cd34 accompanied with other MSC1 marker genes such as Atf3 and Socs3 may promote the generation of MSC2 cells. We speculate that those cells contribute to the genesis of other stromal cell types including endothelial cells. Previous work showed that adipocytes in white and brown fat depots originate from cells that display endothelial characteristics (Gupta et al, 2012; Tran et al, 2012). Activation of PPARγ leads to progressive loss of endothelial cell characteristics and acquisition of an adipocyte phenotype. MSC2 population expressed adipocytes markers such as Fabp4 and Mgll, suggesting that those cells can give rise to adipocytes under the appropriate signaling. Cells that co-expressed Pdgfrβ and Cd34 without Pdgfrα were few in number and lacked adipogenic capacity. We found that MSC1 population expressed numerous ECM genes when the mice were challenged with HFD, and expressed many markers overlapping with fibroinflammatory precursors (FIPs) described by the Gupta group (Hepler et al, 2018). However, Merrick et al (2019) report that in inguinal depots, the MSC1 population did not exhibit a strong inflammatory component (Merrick et al, 2019). This divergence likely stems from the low abundance of those cells that were enriched using Pdgfrβ mural chaser mice (Hepler et al, 2018).

Our computational analysis of cellular crosstalk in fibrotic stroma identified several secreted ligands for receptors on the MSC1 population. Because we focused on alerted gene expression of secreted proteins altered by diet, other non-affected proteins could be expressed in stroma and were not identified by our

approach. We identified several senescence-associated factors, including Opn, TGFβ1, BMP2, TNF, and IL1β in mice subjected to HFD, which are produced by a subpopulation of Cd9+ macrophages. The elevated expression of these factors could directly promote ECM deposition through the activation of the MSC1 progenitor population. We focused our analysis on Opn because it was the only one specifically up-regulated in macrophages by HFD. These findings are consistent with previous reports, which also identified elevated Opn expression by a subpopulation of Cd9+ macrophages in mice subjected to a HFD (Jaitin et al, 2019; Sárvári et al, 2021). We found that OPN enhanced MSC1 proliferation and led to increased expression of ECM genes. Interestingly, Opn deficiency in mice protects against the metabolic complications of diet-induced obesity by reducing the inflammatory status of WAT (Nomiyama et al, 2007). Furthermore, progenitor cells from Opn-deficient mice display an enhanced capacity for differentiating to adipocytes (Moreno-Viedma et al, 2018). Lee et al (2013), identified Opn to be up-regulated by macrophages in inguinal depot after β3-adrenergic receptor (Adrbr3) agonist stimulation (Lee et al, 2013). In concordance with our findings, these authors showed that Opn attracted and induced progenitor cells proliferation. Although Opn promotion of progenitor cell proliferation seems to occur in WAT remodeling induced by both Adrbr3 agonist and high-fat feeding, previous single-cell RNA seq in the browning context did not identify an analogous population of CD9⁺ senescent macrophages (Hepler et al, 2018; Rajbhandari et al, 2019; Rabhi et al, 2020).

In conclusion, we show that obesity induces macrophage senescence and dysfunction. Those cells express a new set of

profibrotic secreted factors including OPN, which promote early progenitor cells proliferation and ECM production. Further studies are needed to identify the mechanisms controlling macrophages phenotypic shift and its impact on the overall stroma microenvironment. Macrophages respond to early signs of tissue damage or invading organisms and are poised to stimulate other immune cells to respond when danger signals are phagocytosed (Murray & Wynn, 2011). The senescent phenotype promoted by obesity may alter their ability to restore homeostasis after an injury or an inflammatory reaction and could promote a favorable environment for chronic diseases. In line with this concept, OPN a major protein we found secreted by CD9[+] senescent macrophages in obese state has been shown to delay resolution of liver fibrosis (Leung et al, 2013). Furthermore, immune cells senescence including macrophages has been implicated in pathogenesis of severe age-related diseases such as Alzheimer's, diabetes, osteoporosis, atherosclerosis, and cancer (Licastro et al, 2005; Fulop et al, 2010; Jackaman et al, 2017). Our observations may explain the striking similarities between age- and obesity-related pathologies, and provide a conceptual framework for the discovery of rational therapeutic targets for obesity complications.

# Materials and Methods

### Animals

C57BL/6J mice were purchased from The Jackson Laboratory (Jax) and acclimated for 2–5 wk before studies. Mice were housed in a temperature-controlled environment with a 12-h light–dark cycle with ad libitum water and standard chow diet. All animal studies were approved by the Boston University School of Medicine Institutional Animal Care and Use Committee.

### Isolation of eWAT SVCs

Epididymal fat pads were isolated from 24-wk-old, male C57BL/6J mice (#000664; Jax) using the mouse adipose tissue dissociation kit (Miltenyi Biotec). For cell culture, cells were seeded with primary cell culture medium (DMEM/Ham's F-12 with 10% FBS and 1% pen strep) in wells pre-coated with 1% Matrigel hESC-Qualified Matrix (#354277; Corning). The medium was replenished every 2–3 d. For adipogenesis, cells were incubated in induction medium containing 1 $\mu$M dexamethasone, 870 nM insulin, 0.5 mM IBMX, and 1 $\mu$M rosiglitazone for 2 d. Cells were maintained in differentiation medium containing the same concentrations of insulin afterwards.

### Proliferation assays

Isolated SVC was resuspended in PBS containing 0.5% (wt/vol) BSA and 2 mM EDTA. Sca1[+] cells were enriched after CD45[+] cell depletion using Miltenyi beads according to the manufacturer's instructions. Isolated cells were seeded at 1,000 cells per well into black, flat-bottomed 96-well plates (#3603; Corning). After 24 h, adherent cells were treated with culture medium containing 3 $\mu$g/ml OPN (#120-35; Peprotech), 10 ng/ml PDGF-BB (#100-14B; Peprotech), or both for 24,

48, 72, and 96 h. Cells were washed with PBS and fixed in 4% PFA for 15 min at after treatments. Nuclei were then stained with 1 $\mu$g/ml Hoechst 33342 for 10 min, before imaging and counting using Celigo Live Imaging Cytometer (Nexcelom Bioscience).

### FACS

Freshly isolated eWAT SVC were resuspended in FACS buffer (PBS/1% BSA). Samples were blocked with mouse Fc block (1:50; BioLegend) for 5 min and then incubated with antibody mix supplemented brilliant stain buffer (BD Biosciences) and monocyte blocker (BioLegend) for 20 min at 4°C protected from light. The SVC suspension was rinsed three times before sorting and flow cytometry analysis with using BD FACSAria II Cell Sorter (BD Biosciences). Frequencies of each cell type were calculated using FlowJo and data were plotted and compared using Prism 6.0 (GraphPad).

### Phagocytosis assay

Freshly isolated eWAT SVC from 25-wk LFD or HFD feed 20 C57BL/6J were resuspended in DMEM/Ham's F-12 with 10% FBS and 1% pen strep after red blood cell lysis, then incubated with PC/PS Lipid Microparticles (# P-B1PCPS-2; Echelon Biosciences), at 1:200 ratio for 30 min at 37°C. Cells were then centrifuged at 300$g$ for 10 min, washed with FACS Buffer before antibodies staining. Macrophages populations phagocytic capacity was then analyzed in BD FACSAria II Cell Sorter (BD Biosciences). Frequencies of each cell type were calculated using FlowJo and data were plotted and compared using Prism 6.0 (Graphpad).

### Immunofluorescence staining

Paraffin-embedded inguinal adipose tissue 5 $\mu$m sections were mounted onto slides, deparaffinized, and rehydrated before performing antigen retrieval. Tissue sections were stained with antibodies overnight at 4°C. After washing with 0.1% tween-20 TBS, the sections were incubated for 1 h at room temperature with fluorophore conjugated secondary antibody. Slides were then washed three times with 0.1% tween-20 TBS at room temperature in the dark. Coverslips were mounted using Prolong gold antifade (Thermo Fisher Scientific). Fluorescent images for all stained adipose tissue sections were captured with an Axio scan Z1 imager (Zeiss) at 20× magnification.

### X-gal staining

Freshly isolated epididymal fat pads were cut into 50-mm chunks and fixed in 2% PFA and 0.5% glutaraldehyde for 15 min. Fixed tissues were washed three times in PBS for 15 min each. X-gal staining was performed by incubating tissues in X-gal working solution (#745-740; Boehringer-Mannheim) overnight at 37°C. X-gal working solution was prepared according to the manufacturer's instructions. The following day, stained tissues were washed in PBS for 5 min each then imaged.

### Oil Red O staining

Cultured cells were fixed with 4% PFA for 20 min then washed briefly with PBS. Oil Red O staining and quantification of the staining intensity were performed using the Lipid (Oil Red O) Staining Kit (#K580; BioVision) according to the manufacturer's instructions.

### Real-time PCR

Total RNA was extracted from frozen tissues and cells using TRIzol reagent according to the manufacturer's instructions. RNA concentrations were determined on NanoDrop spectrophotometer. Total RNA (100 ng to 1 $\mu$g) was transcribed to cDNA using Maxima cDNA synthesis (Thermo Fisher Scientific). Quantitative real-time PCR was performed on ABI Via detection system, and relative mRNA levels were calculated using comparative threshold cycle (CT) method. SYBR green primers are listed in Table S2.

### Single-cell RNA sequencing

Cells were prepared for single-cell sequencing according to the 10× Genomics protocols. Sequencing was performed on Illumina Next-Seq500. About 1,120 cells were captured for each condition with mean reads per cell of 20,707. The Cell Ranger Single-Cell Software Suite (v.3.1.0) (available at https://support.10xgenomics.com/single-cell-gene-expression/software/pipelines/latest/what-is-cell-ranger) was used to perform sample demultiplexing, barcode processing, single-cell 3′ counting, and counts alignment tomm10 mouse reference genome. For further analysis, the R (v.3.1) package Seurat was used (adapted workflow available at https://satijalab.org/seurat/v3.1/immune_alignment.html) (Satija et al, 2015; Stuart et al, 2019). Briefly, cells with feature counts more than 2,500 or less than 200 or have more than5% of mitochondrial genes were filtered out. All the samples were integrated and top 40 dimensions were used to generate the final clusters. To distinguish between stem and progenitor cells, the difference between the G2M and S phase scores was regressed out. Cells are represented with t-distributed stochastic neighbor embedding (t-SNE) plots. The Seurat function "FindNeighbors" followed by the function "FindClusters" were used for clustering using resolution of 0.9. FindAllMarkers function was used to identify specific gene markers for each cluster. Violin plots were used to compare selected gene expression. Differential expression between clusters was obtained using MAST. Specific genes for each cluster were used for functional annotation and Go terms using fgsea and pseudotime was analyzed using Slingshot (Street et al, 2018).

### Whole-mount tissue processing, clearing, and imaging

Freshly isolated epididymal fat pads were cut into small pieces then fixed in 4% PFA/PBS at 4°C overnight. After permeabilization for 18 h in 0.3% (vol/vol) Triton X-100 PBS at 4°C, the samples were blocked with 5% (vol/vol) donkey serum in 0.05% (vol/vol) Triton X-100 PBS for 12 h at 4°C and then incubated with the primary antibody for 12 h at 4°C. For OPN staining, an additional incubation with a fluorophore conjugated secondary antibody was carried out in 0.05% (vol/vol) Triton X-100 PBS for 12 h at 4°C. Samples were then washed in 0.05% (vol/vol) Triton X-100 PBS at room temperature for 4 h.

DAPI (0.1 $\mu$g/ml) was add in PBS for 4 h. Samples were subjected to a second round of fixation before performing washes in gradients (25%, 50%, 65%, 85%, 90%, 95%, and 100%) of methanol for 1 h each at room temperature followed by a wash in a series (50%, 75%, 100%) of benzoic acid:benzyl benzoate (Sigma-Aldrich) in a 1:2 ratio before being imaged in 3D on a LSM 880 (Zeiss) using Fast Airy scan mode.

### Statistical analysis

Data were analyzed using GraphPad Prism software (GraphPad) or R software and are presented as mean ± SEM. Statistical significance was 20 determined by unpaired two-tailed $t$ test or ANOVA; a $P$-value of ≤0.05 was considered significant.

## Data and Materials Availability

All data are available in the main text or the supplementary materials. Single-cell data sets generated during this study y are available at GEO accession GSE194399.

## Supplementary Information

## Acknowledgements

We thank Hu Tianmu and Yuriy Alekseyev of the Boston University School of Medicine (BUSM) Single Cell Sequencing Core for their advice and assistance. We also thank the BUSM Flow Cytometry Core Facility for support. This work was supported by the National Institutes of Health/National Institute of Diabetes and Digestive and Kidney Diseases (DK117161 and DK117163 to SR Farmer), the American Heart Association fellowship (17POST33660875 to N Rabhi), and The Evans Center for Interdisciplinary Biomedical Research ARC on "Connecting Tissues and Investigators, Fibrosis in Pathology" at Boston University (MA).

### Author Contributions

N Rabhi: conceptualization, data curation, funding acquisition, and writing—original draft, review, and editing.
K Desevin: investigation.
AC Belkina: resources, investigation, and writing—review and editing.
A Tilston-Lunel: investigation.
X Varelas: supervision and funding acquisition.
MD Layne: conceptualization, funding acquisition, and writing—review and editing.
SR Farmer: conceptualization, funding acquisition, and writing—review and editing.

### Conflict of Interest Statement

The authors declare that they have no conflict of interest.

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
