## [Reviewer comments · Life Science Alliance]

Life Science Alliance

Obesity-induced senescent macrophages activate fibrogenic gene expression in adipocyte progenitors

Stephen R. Farmer, Nabil Rabhi, Kathleen Desevin, Anna Belkina, Andrew Tilston-Lunel, Xaralabos Varelas, and Matthew Layne

DOI: <https://doi.org/10.26508/lsa.202101286>

Corresponding author(s): Stephen R. Farmer, Boston University and Matthew Layne, Boston University

Review Timeline:

Submission Date:	2021-11-02
Editorial Decision:	2021-11-03
Revision Received:	2022-01-19
Editorial Decision:	2022-01-21
Revision Received:	2022-01-27
Accepted:	2022-01-31

Transaction Report:

Please note that the manuscript was previously reviewed at another journal and the reports were taken into account in the decision-making process at *Life Science Alliance*. Since the original reviews are not subject to Life Science Alliance's transparent review process policy, the reports and author response cannot be published.

November 3, 2021

Re: Life Science Alliance manuscript #LSA-2021-01286-T

Prof. Stephen R. R Farmer
Boston University
Dept. of Biochemistry
Boston, 715 Albany Street USA-Boston, MA 02118

Dear Dr. Farmer,

Thank you for submitting your manuscript entitled "Obesity-induced senescent macrophages activate a fibrotic transcriptional program in adipocyte progenitors" to Life Science Alliance. We invite you to re-submit the manuscript, revised to address the following:

- Address Reviewer 1's comments
- Address Reviewer 2's comments

Thank you for this interesting contribution to Life Science Alliance. We are looking forward to receiving your revised manuscript.

Sincerely,

Eric Sawey, PhD
Executive Editor
Life Science Alliance
<http://www.lsa-journal.org>

- A letter addressing the reviewers' comments point by point.
- An editable version of the final text (.DOC or .DOCX) is needed for copyediting (no PDFs).
- High-resolution figure, supplementary figure and video files uploaded as individual files: See our detailed guidelines for preparing your production-ready images, <https://www.life-science-alliance.org/authors>
- Summary blurb (enter in submission system): A short text summarizing in a single sentence the study (max. 200 characters including spaces). This text is used in conjunction with the titles of papers, hence should be informative and complementary to the title and running title. It should describe the context and significance of the findings for a general readership; it should be written in the present tense and refer to the work in the third person. Author names should not be mentioned.
- By submitting a revision, you attest that you are aware of our payment policies found here: <https://www.life-science-alliance.org/copyright-license-fee>

B. MANUSCRIPT ORGANIZATION AND FORMATTING:

January 21, 2022

RE: Life Science Alliance Manuscript #LSA-2021-01286-TR

Prof. Stephen R. R Farmer
Boston University
Dept. of Biochemistry
Boston University School of Medicine
72 E Concord Street
Boston, Massachusetts USA-Boston, MA 02118

Dear Dr. Farmer,

Thank you for submitting your revised manuscript entitled "Obesity-induced senescent macrophages activate fibrogenic gene expression in adipocyte progenitors". We would be happy to publish your paper in Life Science Alliance pending final revisions necessary to meet our formatting guidelines.

- please upload your Tables in editable .doc or excel format;
- please add ORCID ID for secondary corresponding author-they should have received instructions on how to do so
- please consult our manuscript preparation guidelines <https://www.life-science-alliance.org/manuscript-prep> and make sure your manuscript sections are in the correct order and labeled correctly
- please add your main, supplementary figure, and table legends to the main manuscript text after the references section
- please indicate scale bar size in Legends for figure 3B, 4E, 6B, 6C, and 7D
- Please upload all figure files as individual ones, including the supplementary figure files; all figure legends should only appear in the main manuscript file
- please remove label A in the figure which consists of only one panel (S7), and correct its legend and callout in the manuscript text
- please include the GEO accession number for the single cell data in the Data Availability statement

FIGURE CHECKS:

- scale bars for figure 3B, 4E, 6B, C, and 7D are hardly visible

A. FINAL FILES:

B. MANUSCRIPT ORGANIZATION AND FORMATTING:

Sincerely,

January 31, 2022

RE: Life Science Alliance Manuscript #LSA-2021-01286-TRR

Prof. Stephen R. R Farmer
Boston University
Dept. of Biochemistry
Boston University School of Medicine
72 E Concord Street
Boston, Massachusetts USA-Boston, MA 02118

Dear Dr. Farmer,

Thank you for submitting your Research Article entitled "Obesity-induced senescent macrophages activate fibrogenic gene expression in adipocyte progenitors". It is a pleasure to let you know that your manuscript is now accepted for publication in Life Science Alliance. Congratulations on this interesting work.

DISTRIBUTION OF MATERIALS:

Again, congratulations on a very nice paper. I hope you found the review process to be constructive and are pleased with how the manuscript was handled editorially. We look forward to future exciting submissions from your lab.

Sincerely,
